# The global scientific research response to the public health emergency of Zika virus infection

**Juliane Fonseca de Oliveira**[1,2☯]*, **Julia Moreira Pescarini**[1☯], **Moreno de Souza Rodrigues**[1,3☯], **Bethania de Araujo Almeida**[1☯], **Claudio Maierovitch Pessanha Henriques**[4☯], **Fabio Castro Gouveia**[5☯], **Elaine Teixeira Rabello**[6,8☯], **Gustavo Correa Matta**[7☯], **Mauricio L. Barreto**[1,9☯], **Ricardo Barros Sampaio**[1,4☯]*

**1** Centro de Integração de Dados e Conhecimentos para Saúde (Cidacs), Instituto Gonçalo Moniz, Fiocruz, Salvador, Bahia, Brazil, **2** Centro de Matemática da University of Porto, Porto, Portugal, **3** Laboratório de Análise e Visualização de dados, Fiocruz, Rondônia, Brazil, **4** Gerência Regional de Brasília, Fiocruz, Brasília, Brazil, **5** Casa Oswaldo Cruz, Fiocruz, Rio de Janeiro, Brazil, **6** Instituto de Medicina Social, Universidade do Estado do Rio de Janeiro, Rio de Janeiro, Brazil, **7** Escola Nacional de Saúde Pública, Fiocruz, Rio de Janeiro, Brazil, **8** Public Administration and Policy Group, Wageningen University & Research, Wageningen, The Netherlands, **9** Instituto de Saúde Coletiva, Universidade Federal da Bahia, Salvador, Bahia, Brazil

☯ These authors contributed equally to this work.
* rsampaio.br@gmail.com (RBS); julianlanzin@gmail.com (JFO)

**Data Availability Statement:** All relevant data are within the manuscript and its Supporting Information files.

**Funding:** This work is supported by the Secretary of Health Surveillance, Brazilian Ministry of Health,

## Abstract

### Background

Science studies have been a field of research for different knowledge areas, and they have been successfully used to analyse the construction of scientific knowledge, practice and dissemination. In this study, we aimed to verify how the Zika epidemic has moulded the scientific articles published worldwide by analysing international collaborations and the knowledge landscape through time, as well as research topics and country involvement.

### Methodology

We searched the Web of Science (WoS), Scopus and PubMed for studies published up to 31st December 2018 on Zika using the search terms "zika", "zkv" or "zikv". We analysed the scientific production regarding which countries have published the most, on which topics, as well as country level collaboration. We performed a scientometric analysis of research on Zika focusing on knowledge mapping and the scientific research path over time and space.

### Findings

We found two well defined research areas divided into three subtopics accounting for six clusters. With regard to country analysis, the USA and Brazil were the countries with the highest numbers of publications on Zika. China entered as a new player focusing on specific research areas. When we took into consideration the epidemics and reported cases, Brazil and France were the leading research countries on related topics. As for international collaboration, the USA followed by England and France stand out as the main hubs. The research

through the Zika Platform - a long-term surveillance platform for the Zika virus and microcephaly in Brazil, at the Center of Data and Knowledge Integration for Health (CIDACS-Fiocruz) This work was partially supported by the European Union's Horizon 2020 Research and Innovation Programme under ZIKAlliance Grant Agreement no. 734548, and by the Oswaldo Cruz Foundation/ Vice-Presidency of Research and Biological Collections-Fiocruz/ VPPCB and the Newton Fund/ British Council.

**Competing interests:** The authors have declared that no competing interests exist.

areas most published included public health-related topics from 2015 until the very beginning of 2016, followed by an increase in topics related to the clinical aspects of the disease in 2016 and the emergence of laboratory research in 2017/2018.

## Conclusions

Mapping the response to Zika, a public health emergency, demonstrated a clear pattern of the participation of countries in the scientific advances. The pattern of knowledge production found in this study represented varying country perspectives, research capacity and interests based first on their level of exposure to the epidemic and second on their financial positions regarding science.

## Introduction

Zika virus infection (ZVI) was first reported in humans in 1954, but only in 2007 was the first-ever outbreak reported in Micronesia [1, 2]. Several small outbreaks in Micronesia and the Pacific Islands have been reported since then, and then ZVI started revealing its epidemic potential [3]. In 2015, the virus was associated with neurological disorders, including Guillain-Barré syndrome, microcephaly and still unknown abnormalities in newborns infants [4, 5]. It was and yet remains an event surrounded by many scientific uncertainties.

In February 2016, mostly driven by the large numbers of Zika-associated microcephaly cases in Brazil, the World Health Organization (WHO) declared a public health emergency of international concern (PHEIC). Different research groups all over the world, as part of separate and collaborative research networks, made a concerted effort to understand the various factors required to cope with the epidemic of Zika virus infection, transmission, pathogenesis, treatment and control, as well as the social impact of severe forms of the disease [3]. During the outbreak, especially after WHO PHEIC, international funding and fast-tracking procedures for publishing scientific outcomes stimulated scientific research and publications. The fast increase in the number of basic researches on Zika from 2015 to May of 2016 has been attributed mainly to its association with microcephaly [6]. However, from then on, only a few core journals have published on Zika [7].

Since the PHEIC declaration, science studies on ZVI have mostly focused on the scientific networks and their collaboration measured by their co-publications. Some studies have adopted bibliometric or scientometric perspectives, analysing papers and other documents to discuss aspects such as co-authorship, cooperative networks and the innovative potential of the ZVI topic [7–10]. In Maia et al [11], a first study on the scientific networks organized around ZVI and their most influential researchers used Social Network Analysis to map the co-authorship networks of Zika papers published in 2015/2016. The study showed that, besides the number of publications, the prominence of a given researcher in scientific networks can be explained by the diversity of partnerships and the establishment of connections with the pioneers in the field [11]. Also, a recent study on Zika-related publications and patents from a scientometric perspective highlighted the strategic role played by the Latin America and Caribbean research network in terms of addressing the science and technology (S&T) challenges related to the outbreak [12]. The debate raised by such scientometrics studies applied to Zika enriches the discussion about the influence and centrality of certain groups, researchers and even countries.

The bibliometric and scientometric studies have enabled the systematisation of the current state of knowledge, particularly which research areas have been most developed. However, so far, no scientometric study has focused on identifying interactions and collaborations among researchers from different countries to inform the development of shared governance in research-related to ZVI. We aim to study how the 2015–16 Zika epidemic moulded the world-wide scientific production on ZVI. Furthermore, by ranking the top ten countries in publications, we also analysed the collaboration between researchers from countries and the evolving knowledge landscape over time, identifying the most researched areas and countries' contribution. We the results are expected to support funders, policy-makers, and scientific institutions in defining priorities and in allocating resources for research in ZVI.

## Materials and methods

### Data collection, treatment and cleaning

Our data were obtained by searching terms related to Zika virus infection ("zika" OR "zkv" OR "zikv") by Topic in the Web of Science (WoS), PubMed and Scopus. We selected all the articles published from 1945 to the end of December 2018. The search was performed on October 15th, 2019. From the retrieved articles, we extracted the title and abstract to develop a knowledge map. We used the metadata variables regarding date published, authors and co-authors' affiliation countries as well as the ISI subject categories from WoS, assigned to each paper [13].

The selected contents were downloaded as a text file containing all the results of our query. The file was cleaned by (i) converting all words to lowercase, (ii) removing special characters (e.g., ".", ":" and "/"), and (iii) standardizing similar terms (e.g., "ae aegypti", "aedes aegypti"). Data cleaning was conducted in Python. The Python code used is available at GitHub [14].

### Data analysis

In the first subsection of the results section, we conducted a descriptive analysis of the articles retrieved from the three different sources: WoS, Scopus and PubMed. Based on the results found we restricted our further analysis using only WoS articles since the difference in the corpora were quite small compared to the benefits of the categorization and classification methods used by WoS (see S1–S3 Figs). Overall, WoS provides a complete information for each article. For instance, information about authors' institutional affiliation country is not fully present in PubMed. Moreover, WoS provides summarized subject categories, and although Scopus also presents subject categories, they are not aligned with those of WoS. From that on, using only the WoS database, we performed two separate analyses, which we describe below: a descriptive analysis to understand how countries collaborate towards Zika knowledge; and, a textual analysis of research on Zika using knowledge mapping over time and space (i.e., countries). In order to summarize the findings in this manuscript, we present our results for the top ten countries with the most publications.

### Descriptive analysis

The descriptive analysis was performed both directly by filtering information from the Web of Science homepage and by using VOSviewer software [15] to perform country collaboration network analysis. To summarize data information for each research article, we extracted from each author his institutional affiliation country. Articles produced by different institutions in the same country were considered a national collaboration, and articles produced by authors affiliated to institutions from more than one country were considered international collaborations.

After that, we constructed a co-authorship network at the country level using data exported from VOSviewer and reconstructed and filtered using Gephi software for visualization [16].

In order to measure collaboration in Zika publications, we build up an "international collaboration index" (ICI), which is the ratio of the number of countries that a given country collaborated with, over the total number of countries (minus the country itself) which produced results toward Zika knowledge. The closer the index is to one, the more internationally collaborative the country is, and the closer it is to zero, the less collaborative the country is.

To address the issue of international collaboration over time as a proportion of published articles, an international collaboration factor (ICF) was calculated as the number of articles with author(s) affiliation from more than one country over the number of articles with author(s) affiliation to just one country. Similar to the ICI, the more collaborative a country is, the closer it is to one, the less collaborative, the closer it is to zero.

## Textual analysis

We carried out a textual analysis to discover textual data relationships among extracted words. The method reveals thematic trends and identifies publication subjects surrounding specific topics or fields [17]. To do this, we analysed the titles and abstracts of the articles in our sample. More specifically, we analysed words (also called terms or forms), extracted from the titles and abstracts, grouped by text segments comprised of 40 terms each, and applied Correspondence Factorial Analysis (CFA) to associate them. The baseline assessment of research areas into clusters relied on the clustering algorithm of IRAMuTeQ v7.2 (Interface for Multidimensional Analysis of Texts and Questionnaires). The association process is obtained by testing if the frequency of a given word is statistically associated with another using Pearson Chi-square at a significance level of 5%. Based on the similarity, words are aggregated into clusters of main areas of knowledge. Content analysis was also performed using IRAMuTeQ.

The defined number of clusters in IRAMuTeQ required empirical knowledge in the field as well as several tests on how many clusters made a fair and accurate separation of knowledge areas. Therefore, to validate our baseline clusters, we compared the number and content of each cluster obtained using IRAMuTeQ with those obtained using VOSviewer. To associate WoS categories with the abstract clusters defined by IRAMuTeQ, we first selected the ten most frequent categories (that is given by WoS) presented in our studied articles. Second, we analysed if the most significant categories for each one of the 6 clusters independently appeared with a high frequency or not. By doing this, we were able to better classify the clusters by the words which appeared in each one of them. Moreover, we used the categories defined by WoS for each article, based on the journal in which it was published to further evaluate if the clustering process and defined sub-areas of research were aligned. Finally, we showed the results to a panel of 12 specialists on Zika and public health to validate the clusters and their nomenclature.

After we had established statistically significant knowledge area clusters, we evaluated which knowledge areas emerged over time, and stratified them into four groups by publication year: < = 2015, 2016, 2017 or 2018; second, we evaluated the participation in each cluster/field of the top 10 countries with regard to the number of publications.

## Results

### Descriptive analysis of the databases WoS, Scopus and PubMed

The collected data yielded a total of 3,546, 3,793 and 4,835 articles retrieved from the WoS, Scopus, and PubMed databases, respectively. After merging (see S4 Fig), a total of 6,209 articles were unique registered (Fig 1).

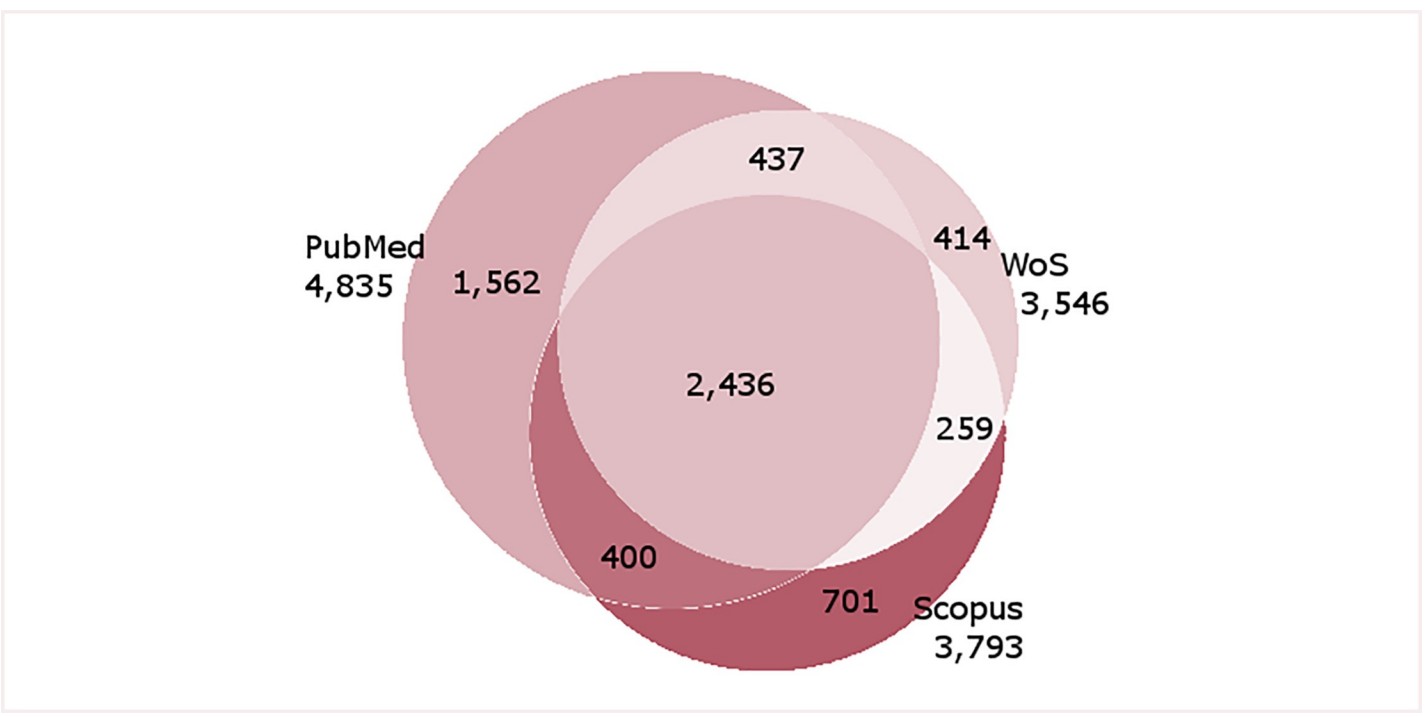

**Fig 1. Articles identified in PubMed, WoS and Scopus database published from 1945 to the end of December 2018.**

## Descriptive analysis of the countries collaborations

The total of 3,546 articles found in WoS were produced by 18,516 authors, associated with institutions from 150 countries. These articles were published with and without international collaboration. Fig 2 shows a dispersion matrix using the number of publications and ICI by country.

According to Fig 2 The most internationally collaborative countries were the USA (ICI = 0.67), followed by France (ICI = 0.57), England (ICI = 0.55), Germany (ICI = 0.47) and Brazil (ICI = 0.45). The remaining most productive countries: Italy, India, Australia, China and Canada, presented international collaboration indexes of 0.41, 0.40, 0.35, 0.32 and 0.31, respectively. When we take into consideration the ICI with the number of publications we can see that the USA has the highest ICI and number of publications. In contrast, China, the third country with the highest number of publications, is the ninth in ICI. Brazil occupies an intermediate position regarding the number of collaborating countries and the number of publications, with the second-largest number of publications on Zika, and the fourth in ICI. France and England seem to be well-positioned in terms of the number of collaborating countries. The remaining five countries have fewer than 200 publications and show a similar ratio between the logarithmic number of publications and the number of countries with which they have collaborated.

Amongst the ten countries that published the most on Zika during the study period, the USA and Brazil followed by China, France and England accounted for almost 88% of the scientific production from WoS (Table 1). On average, international collaborations were present in 61.11% of articles produced by authors from the top 10 countries. The USA was the only country with more than 50% of its articles resulting from no international collaboration (57.22%). In contrast, England has the highest number of publications that resulted from international

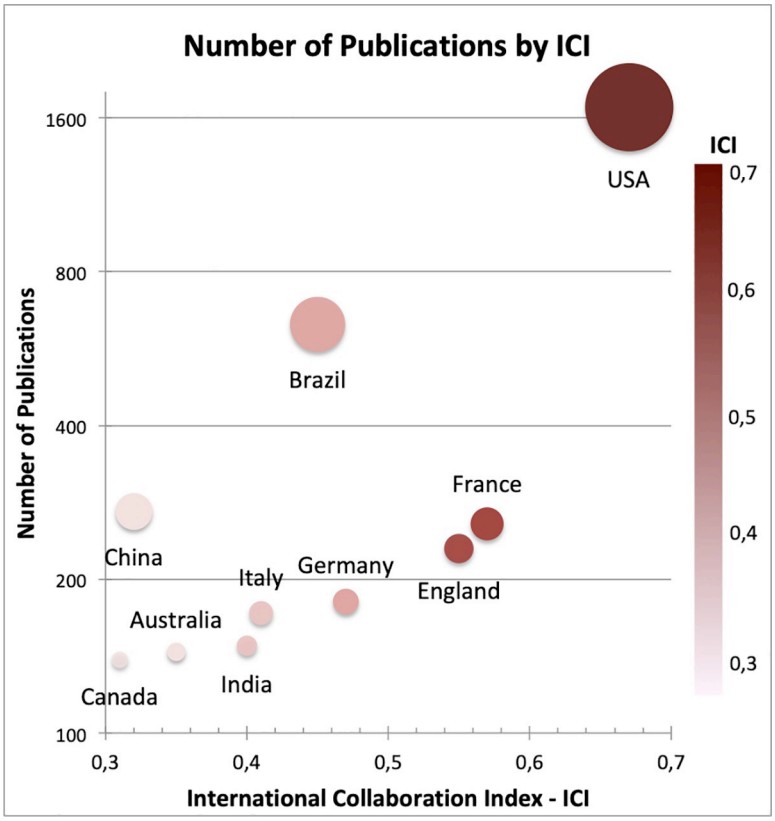

**Fig 2. Number of publications according to the International Collaboration Index for each of the top ten most productive countries.** The graph is a dispersion matrix, the y-axis is the number of publications on a logarithmic scale base 2 starting at 100 publications and the ICI is on the x-axis starting at 0.3. The size of the circles is related to the number of publications, and the intensity of the colour from dark red to light pink is related to the ICI as shown on the right-side scale.

collaboration (84.75%), followed by Italy (75.57%) and Germany (74.73%) (Table 1). Despite being in the 10th position, the number of articles that the USA published with international collaboration corresponds to more than three times the total number of articles published by England, the country with the highest international collaboration ratio.

Among the top 10 countries with the most scientific publications on Zika, the evolution of their respective ICFs over time showed a balanced level of production with and without international collaboration for USA, Brazil, China and Australia (Fig 3). In contrast, England, Germany, France and Italy published articles with international collaborations at a higher frequency during the entire period of analysis.

Using co-authorship networks between countries constructed from VOSviewer, we see that the majority (61.89%) of the articles were produced within a country, 24.17% were collaborations between two of the countries, 8.88% between three countries, 2.51% between four countries and 2.55% between five or more countries.

Out of the 735 articles published by the USA with international collaboration, over 15% were with Brazil and nearly 10% with China (Fig 4). Out of the 341 Brazilian studies published as results of international collaboration, over 60% were with the USA, followed by nearly 20% with England and 10% with both France and Germany. For the 151 Chinese studies published with international collaboration, over 75% were with the USA and a few articles with other countries (Fig 4).

**Table 1. National and international publications about Zika virus infection for the top 10 most productive countries.**

| Countries | Articles Total[*] | | Articles with single country author(s)[**] | | Articles with international collaboration)[**] | |
|---|---|---|---|---|---|---|
| | N | % | N | % | N | % |
| **USA** | 1718 | 48.52 | 983 | 57.22 | 735 | 42.78 |
| **Brazil** | 648 | 18.27 | 307 | 47.38 | 341 | 52.62 |
| **China** | 279 | 7.87 | 128 | 45.88 | 151 | 54.12 |
| **France** | 264 | 7.44 | 70 | 26.52 | 194 | 73.48 |
| **England** | 236 | 6.66 | 36 | 15.25 | 200 | 84.75 |
| **Germany** | 186 | 5.24 | 47 | 25.27 | 139 | 74.73 |
| **Italy** | 176 | 4.98 | 43 | 24.43 | 133 | 75.57 |
| **India** | 151 | 4.28 | 55 | 36.42 | 96 | 63.58 |
| **Australia** | 148 | 4.17 | 67 | 45.27 | 81 | 54.73 |
| **Canada** | 142 | 4.02 | 45 | 31.7 | 97 | 68.3 |

[*] The column "Articles Total" represents the total number of articles published by a country and the percentage of these productions over the total number of articles about Zika found in WoS until 2018.

[**] Among these values, we see the number of articles produced by the country itself and the proportion produced with international collaboration.

## Main research areas of knowledge

A total of 24,685 words were analysed inside 21,055 text segments, of which 17,525 were considered active words (i.e., verbs, nouns and adjectives). Using IRAMuTeQ, two groups (A and B) and six clusters were identified and named according to the most frequent words. Cluster 1 was named "Clinical Aspects" (13.8% of the words); Cluster 2 "Diagnosis" (10.6%); Cluster 3 "Epidemiology" (20.1%); Cluster 4 "Entomology" (14.5%); Cluster 5 "Cellular Biology" (19.9%); and Cluster 6 "Molecular Biology" (21%) (Fig 5).

We identified that 165 of 252 possible WoS categories were present in the studied articles. Each article is classified in one or more categories. The name (words) of these categories also appear distributed in the clusters defined in Fig 5. The ten most frequent categories were: Infectious Diseases; Public Environmental Occupational Health; Tropical Medicine; Virology; Multidisciplinary Sciences; Microbiology; Parasitology; Immunology; Biochemistry Molecular Biology; and Medicine General Internal. Anther six categories, among the 10% most frequent, were not included in the top ten list, but were significant to categorize the clusters, these were: Entomology; Cell Biology; Health Policy and Services; Clinical Neurology; Pediatrics; and Diagnosis. Based on these sixteen categories, we associated WoS categories with the abstract clusters defined by IRAMuTeQ. We found the following results for Group A: Public Environmental Occupational Health category in the "Epidemiology" cluster (cluster 3); Parasitology category in "Entomology" (cluster 4); and Infectious Diseases and Tropical Medicine in both "Diagnosis" (cluster 2) and "Entomology" clusters. Considering that Diagnosis, Epidemiology and Entomology categories from WoS were in cluster 2, 3 and 4, respectively, each cluster was named after these categories. Cluster 1, "Clinical Aspects", had none of the top 10 high-frequency categories of WoS in it.

Analogously, in Group B, with clusters "Cellular Biology" and "Molecular Biology" (cluster 5 and 6), there were similarities in Virology and Microbiology as main categories. The main difference was that in cluster 6, the most common category was Immunology, and in cluster 2 Virology with Multidisciplinary Sciences were some of the areas. The category Cell Biology

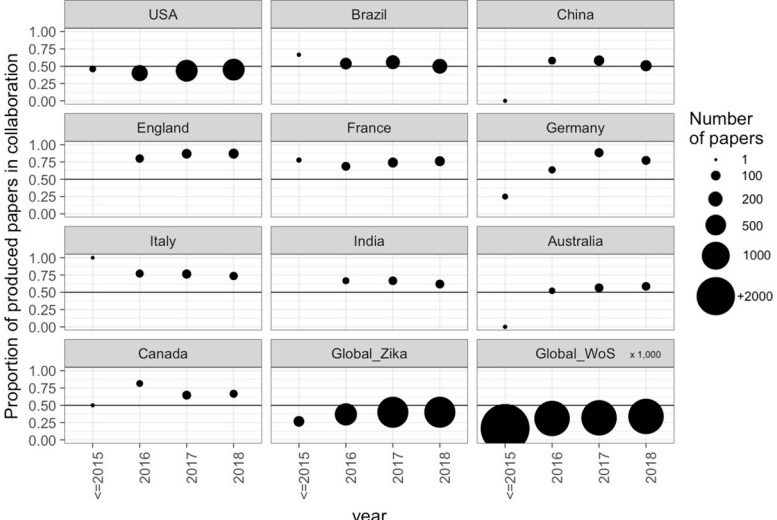

**Fig 3. The evolution of articles produced with scientific collaboration among the top 10 most productive countries on Zika production using only articles found in WoS.** The closer the dots are to 0.5 (highlighted by the black line), the articles produced with collaborations are proportional to the articles produced without collaborations. The value corresponding to the year 2015 represents the sum of all work produced by the selected country up until 2015. Points on the zero line correspond to publications that had no international collaboration. The size of the dots for Global WoS articles are divided by 1,000 in order to fit the scale.

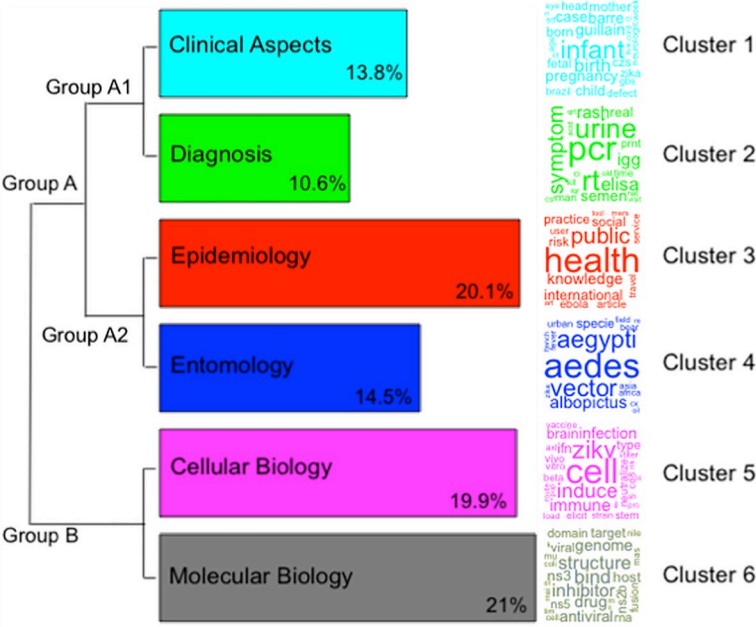

**Fig 4. Collaboration networks among the 10 most productive countries.** Each node corresponds to a country and the size of the node is related to the number of international collaborations they have within that group. The thickness of the lines represents the number of collaborations between any two countries and an approximation of those numbers are presented by the box on the top right. The colour scheme emphasizes link differences between countries since the colour of a line is a mix of the colour of the nodes it connects. The node distribution in the figure has been chosen only to facilitate visualization. The two countries with the fewest number of publications are positioned at the centre of the graph and the most productive to the least from top to bottom and left to right.

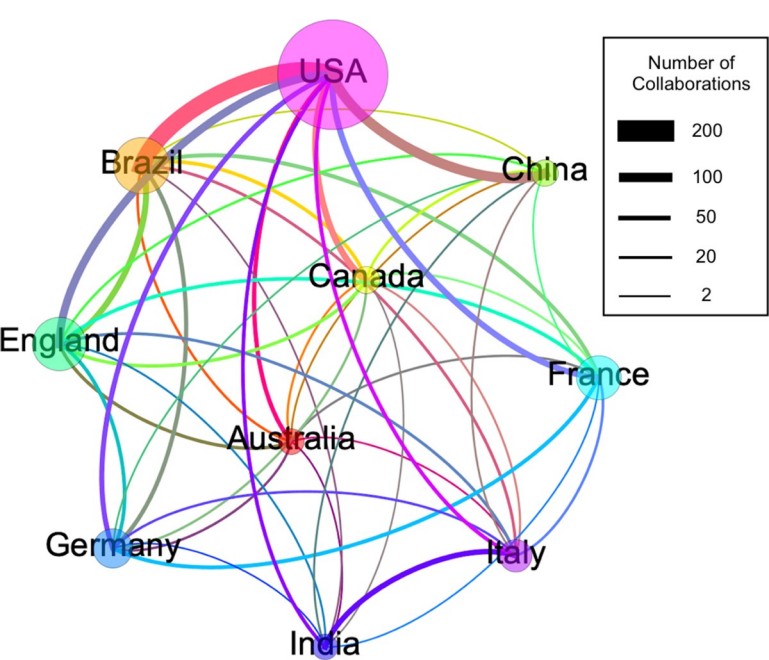

**Fig 5. Dendrogram of the clusters obtained with IRAMuTeQ.** The dendrogram is divided into 6 clusters formed of branch divisions. The first branch divides Group A, and Group B. Group A is further divided into two subgroups called Group A1 and Group A2. The size of each bar corresponds to the percentage of words in that cluster. The associated name for each cluster appears on its bar, and the colour scheme has been chosen to facilitate visualization. In front of each bar, there is a square word cloud of words with the highest chi-square value to aid the understanding and the representation of each cluster.

was included in cluster 5, which was named after it. Similarly, the category Biochemistry Molecular Biology was very evident in cluster 6.

After the associations of the clusters by the WoS categories and their representations, we asked a total of 12 specialists to evaluate the clustering results and the nomenclature. All responses were positive in considering the clusters a well-defined set of research sub-areas within the field of Zika taking into consideration the words available and the categories. Spatial distribution of the words coloured according to the cluster was presented to the specialists with the inferred nomenclature and positive responses were received.

The presence of each clustered research area by country and its contribution to research topics are shown in Fig 6. Although the USA, the most productive country on Zika, has a large presence in all the 6 clusters, the most significant presence is on "Molecular Biology", and its least significant presence in "Clinical Aspects" and "Entomology" (Fig 6). Brazil, the country with the second-highest scientific production, has mainly focused its publications on "Clinical Aspects" of the disease, while it is almost absent in "Cellular Biology". On the other hand, China is present in two clusters mainly: "Cellular Biology" and "Molecular Biology". While France is present in "Clinical Aspects", "Diagnosis" and "Entomology", England showed a relevant number of studies published in "Epidemiology" and "Entomology".

## Year and country distribution of areas of knowledge

In the factorial analysis, the studied publications were distributed into four temporal groups: < = 2015 (3.2%), 2016 (20.2%), 2017 (37.0%) and 2018 (39.6%). In Fig 7 we see that publications up to 2015 were mostly related to entomology research, those published in 2015 up to 2016 were related to public health. For 2015, we observed a higher than expected presence of

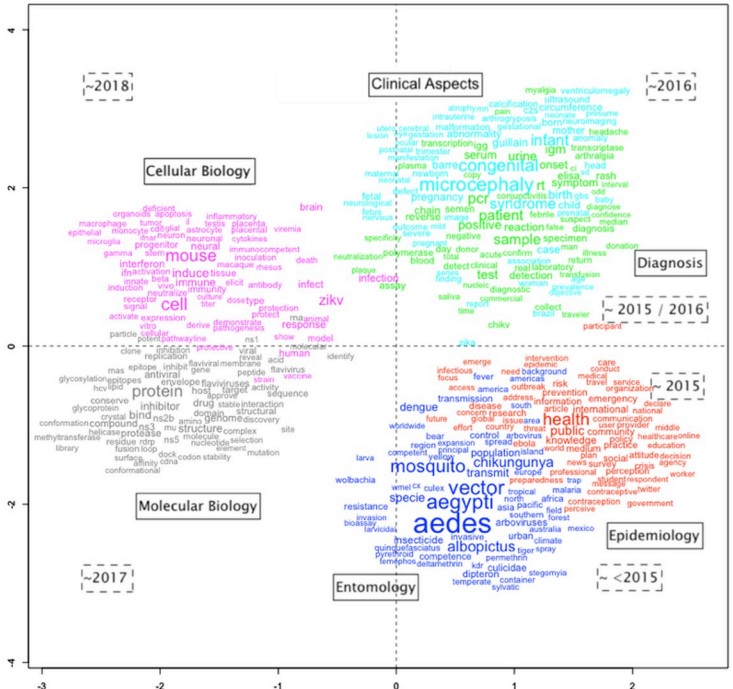

**Fig 6. Chi values for the top 10 countries for each cluster.** The results are calculated taking the observed value (number of text segments) within a cluster minus the expected value for that cluster and country divided by the square root of the expected value. This is the Chi value (not squared) to show negative (or less than expected) values. For each cluster (1 through 6) we have all ten countries and their respective chi value.

the cluster 3 "Epidemiology". Publications related to "Diagnosis" (cluster 2) surged between 2015 and 2016. The top right quadrant has publications from 2016 mostly, where the clusters "Diagnosis" and "Clinical Aspects" are present. The clusters from Group B, identified as the areas of research "Molecular Biology" and "Cellular Biology", appeared only in 2017 and 2018, respectively.

## Discussion

This study shows how ZVI related-science has evolved since 2015 as a result of the interest of an international research community focused on solving new and challenging research questions related to a recognized worldwide health problem. This process mobilized scientific efforts and resources from different countries, institutions and research teams from different scientific areas. In a very short time, complex interactions among these teams from different scientific areas were formed., Using the articles published in this period, our study provides important insights to understand the path of the scientific response to ZVI.

The top ten countries in terms of the number of publications on Zika found in our study is corroborated by the ranking of the top ten countries for scientific research as the largest contributors to papers in leading journals tracked by Nature Index in 2018, as seven of them are in that list [18]. The International Collaboration Index (ICI) and the International Collaboration Factor (ICF) helped us to gain insight into collaborations between countries, and to identify which countries have collaborated more extensively over time. In general, collaborations in science occur through the need for complementary resources among different actors, institutions and countries, such as knowledge and skills, financial resources and access to the field and key actors in order to conduct relevant research. Co-authorship studies indicate that more

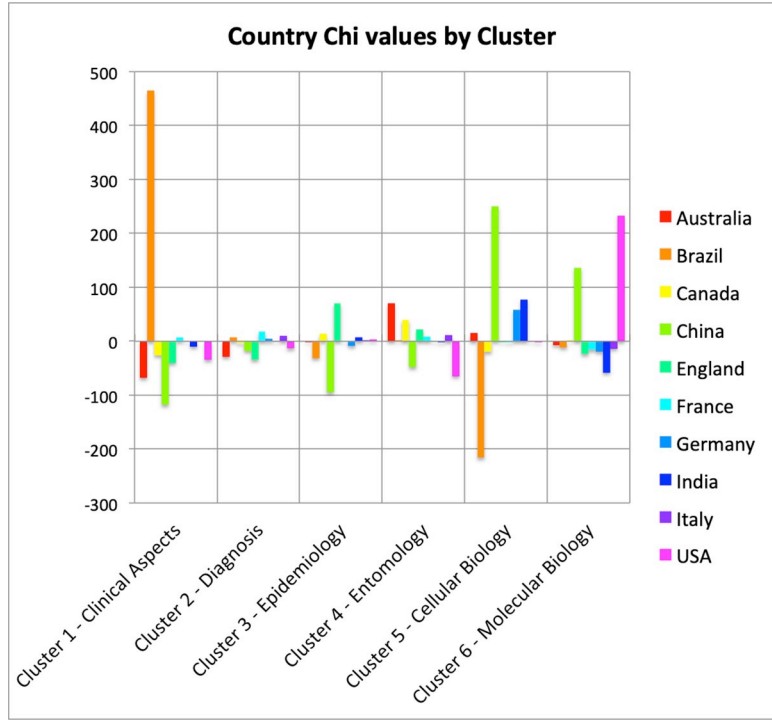

**Fig 7. Knowledge map of words extracted from titles and abstracts of articles on WoS related to Zika virus infection from January of 1945 through December 2018.** The names in the solid line boxes represent the cluster names and the dotted line boxes represent the most frequent periods for the articles represented by the clusters. The size of each word represents their chi-square value; the distance between them represents how statistically related they are based on the results of the correspondence factorial analysis.

connected actors tend to attract a higher number of connections over time [19], an aspect that amplifies learning capacities and the absorption of knowledge.

The majority of scientific production on Zika virus infection was produced by national efforts. On the other hand, international collaborations were present on average in more than 50% of the articles produced by the top 10 countries. The reason for that is the fact that an international paper will account as collaboration for every country represented by that paper but only once when comparing articles with and without international collaboration.

The most internationally collaborative countries were the USA, followed by France, England, Germany and Brazil. The intensity of collaboration between the USA and countries on other continents, especially Brazil and China, indicates that the USA is an influential country in terms of mobilizing collaborations, as well as the extent of its research capacity and interest in Zika. Among the possible reasons why the USA is such a prominent publication country and a central player in terms of international collaboration in Zika could be the fact that it is the top R&D funder for neglected diseases [20] and that several cases of Zika virus transmission were reported in 2016–17 in Florida and Texas [21]. Furthermore, this country has considerable knowledge accumulation in tropical diseases, including the Department of Defence, which has been one of the main supporters of scientific research in this area [22, 23].

Brazil ranks fourth among the most internationally collaborative countries with the USA by far the main collaborator followed by England, France and Germany. We believe that the position of Brazil is due to the country's role as the epicenter of the most recent ZVI epidemic, which boosted foreign scientific interest and made it a key partner in scientific publications focused on this topic. Also, the potential threat of a Zika epidemic could partially explain the

increased country collaboration over time. When the first case of Zika virus infection was confirmed in the Northeast region of Brazil, in May 2015, an epidemiological alert by the Pan American Health Organization (PAHO)/WHO was released. The alert recommended Member States to establish and maintain a capacity for ZVI detection, clinical management and an effective public communication strategy [24]. In November 2015, Brazil declared ZVI a national public health emergency and three months later, the WHO declared it a PHEIC [25]. We suggest that a causal effect between the alert/emergency and the increased number of collaborations took place during this period considering the knowledge gaps in understanding ZVI. Brazil is a country with a vast scientific literature on dengue virus transmission, among other arboviruses. However, the results indicate that it had carried out almost no research on Zika virus before the start of the epidemic.

The most collaborative countries in relative terms publications that resulted from an international collaboration were in Europe (England, Italy and Germany). The high levels of the international collaboration of England may be explained by the significant recent investment made available by the British government and philanthropic funding agencies for the research community aiming to respond to the challenges posed by ZVI [26]. The same can be said for the European Union, which counts on four large research consortia (ZikaPlan, ZikAction, ZikAlliance and ZikaVax) coordinated by European organizations in association with research institutions from different regions across the world [27].

The appearance of the ZVI in its epidemic form in the Americas in 2015, particularly in Brazil, initially triggered the development of epidemiological studies in an attempt to observe and understand the disease incidence and its associations as well as studies focused on characteristics of its clinical manifestations. Entomology along with diagnosis were also important areas of research as Zika virus infection is an arboviral disease with similar epidemiology and symptoms of other arboviral diseases transmitted by Aedes mosquitoes, which requires reliable point-of-care diagnosis. In the following years, from 2017 to 2018, the dominance was on laboratory-based studies, with emphasis on molecular and cellular biology.

At the end of 2015, Zika infection and especially microcephaly appeared as a new problem to be tackled. Questions regarding the clinical aspects and epidemiology were the main focus at this moment. Physicians might argue: "What is the Diagnosis?", "What are the physical pathological mechanisms of the virus/disease?", "What is the case definition?", putting the focus on physiopathology [28, 29]. These questions boosted basic science, since research on the clinical aspects and epidemiology depends basically on people and methods, while applied science builds upon cellular and molecular biology, depending on expensive equipment and other technologies and qualifications. This scenario gives an academic advantage to countries with a higher technological density. We saw Cluster 1 "Clinical Aspects" in 2016 being superseded by Clusters 5 and 6 ("Cellular Biology" and "Molecular Biology") in the following years. It also gives clues to understanding the dominant participation of countries such as the USA and China in the clusters related to cellular and molecular biology. Also, French Polynesia and consequently France played a key role as a research centre for public health during the first signs of Zika endemic.

Our results are also subject to limitations as our main source of information was the Web of Science, which have excluded articles not indexed in this database. However, the Web of Science brings advantages in the textual analysis, while Pubmed and Scopus databases ended up having a little impact on consolidated data. Our strategy was chosen as a proxy representing scientific outputs on Zika virus infection. All in all, this study, with its strengths and limitations, shows that the ten most productive countries in terms of scientific publications are predominantly developed countries, with the exception of three developing countries considered emerging economies where health is an important area of research capacity building: Brazil,

China and India. As research and industrial innovation capacity are concentrated in major economies, influence and centrality tend to be more pronounced in comparison to others due to the availability of funding, specialized personnel and technological infrastructure.

These aspects are important when considering the participation of the top ten countries in the research areas of knowledge identified, it seems that publications from countries with lower research capacity are more focused on areas related to clinical, diagnostic, entomologic and epidemiologic aspects. Starting in 2017, the areas of molecular and cellular biology began to emerge as knowledge areas necessary for the development of technologies for prevention, control and treatment, which coincided with the increased presence of countries possessing greater R&D capacity.

The world has seen an increasing number of disease epidemics, especially in low middle-income countries. Ebola, Middle East Respiratory Syndrome and Zika epidemic are more than isolated threats to specific areas and countries, rather they are threats on a global scale. Considering this, we believe that mapping country-specific scientific research areas and international collaborations in response to the Zika virus public health emergency are important as part of epidemic preparedness and response.

Wilder-Smith *et al* [30] highlighted the fact that arboviral research has not been considered an investment research priority over the last five decades. Given the unprecedented number of arboviral disease outbreaks, the threat of yellow fever and the reemergence of dengue and chikungunya and the case of Zika virus infection that moved quickly to a PHEIC, research and interventions focused on arboviruses should be a priority in order to establish effective and sustainable strategies to tackle multiple Aedes-borne arboviral diseases. The authors also recommend that efforts be conducted via new global alliances in order to be more effective.

The case of Zika virus in Brazil received support from the international scientific community in order to build a cooperative network. Multilateral organizations played a key role, particularly the Pan American Health Organization and the World Health Organization that facilitated Brazilian communication with other countries and public health institutions. Facts like these show that countries collaboration reflected countries R&D capacities and efforts to provide knowledge and evidence to answer the epidemic needs related to an emerging disease like Zika, surrounded by scientific uncertainties.

All in all, a broader question remains: what insights did Zika epidemics give to understand the engines of global scientific networks better? We note that international initiatives in response to the official declaration of the Zika virus as a public health emergency were targeted to identify the state of the outbreak, the ongoing research and the potential for collaboration across research groups. Despite the necessity and importance of this approach, objectives, strategies and resources were not aligned, which resulted in some disparities and underestimation of the potential contribution of some countries and networks. In this sense, governance, coordination, integration of efforts and alignment of financing strategies among countries, as a systemic global institutional arrangement (and not a contingency action) are strategic to enhance preparedness for future epidemics. This would strengthen local but networked processes of research, development, as well as technology licensing, production and distribution when available.

## Conclusions

Mapping the response to Zika, a public health emergency, demonstrated a clear pattern of participation by specific countries in terms of scientific advances. The pattern of knowledge production found in this study represented differing perspectives and interests based primarily on a country's level of exposure to the epidemic as well as on funding directed at scientific research.

Brazil has played an important role in the scientific field, especially in clinical studies and epidemiology surveillance that could help to monitor Zika and offer strategic information to international partners and WHO. For Brazilian scientists and their research institutions, Zika outbreak was an opportunity to receive funding for international cooperation and exchange, and to push international publications. During an immense political and economic crisis in Brazil (2014–2018), the Zika outbreak was crucial to financing health research and strengthening health care for the children and families affected.

It is also important to highlight the intense scientific production and cooperation during a short period of time for research and dissemination. This was a result of political commitments to face health emergencies, particularly after the Ebola and during Zika outbreaks which involved funders, national and international ethical committees, and publishers. It is a relevant example to take to other health emergencies and disasters, and rapidly respond to a potential or real humanitarian crisis.

Even though the USA continues to lead scientific production and collaboration, the performance of Brazilian science during and immediately after the Zika outbreak represents a considerable effort to an LMIC to produce high quality of scientific knowledge in spite of economic inequities and lack of infrastructure.

But now, when Zika research funding are not abundant and the urgency of the outbreak was passed, how equitable and sustainable will Zika production and collaboration be for Latin American countries such as Brazil? How can an outbreak improve equity in science and support LMICs to develop and sustain scientific production and technology to respond to specific and general health issues?

We believe that this study contributes to the understanding of the role of collaborations and research coordination efforts focused on local epidemics which have the potential to threaten countries on global scale.

We propose the establishment of common study protocols and governance structures, as well as robust data sharing, intellectual property rights policies and compliance with ethical and regulatory issues to advance practical recommendations that further epidemic preparedness and response.

## Supporting information

**S1 Fig. PubMed.**
(TIF)

**S2 Fig. Scopus.**
(TIF)

**S3 Fig. WoS.**
(TIF)

**S4 Fig. Total articles.**
(TIF)

**S1 Data.**
(ZIP)

**S1 File.**
(DOCX)

## Author Contributions

**Conceptualization:** Mauricio L. Barreto, Ricardo Barros Sampaio.

**Data curation:** Juliane Fonseca de Oliveira, Fabio Castro Gouveia, Elaine Teixeira Rabello, Ricardo Barros Sampaio.

**Formal analysis:** Mauricio L. Barreto.

**Funding acquisition:** Mauricio L. Barreto.

**Investigation:** Juliane Fonseca de Oliveira, Julia Moreira Pescarini, Moreno de Souza Rodrigues, Claudio Maierovitch Pessanha Henriques, Elaine Teixeira Rabello, Gustavo Correa Matta, Mauricio L. Barreto, Ricardo Barros Sampaio.

**Methodology:** Juliane Fonseca de Oliveira, Moreno de Souza Rodrigues, Fabio Castro Gouveia, Elaine Teixeira Rabello, Gustavo Correa Matta, Ricardo Barros Sampaio.

**Project administration:** Mauricio L. Barreto, Ricardo Barros Sampaio.

**Software:** Fabio Castro Gouveia, Ricardo Barros Sampaio.

**Supervision:** Mauricio L. Barreto, Ricardo Barros Sampaio.

**Validation:** Bethania de Araujo Almeida, Claudio Maierovitch Pessanha Henriques, Fabio Castro Gouveia, Elaine Teixeira Rabello, Gustavo Correa Matta, Mauricio L. Barreto.

**Visualization:** Juliane Fonseca de Oliveira, Moreno de Souza Rodrigues, Fabio Castro Gouveia.

**Writing – original draft:** Juliane Fonseca de Oliveira, Julia Moreira Pescarini, Moreno de Souza Rodrigues, Bethania de Araujo Almeida, Claudio Maierovitch Pessanha Henriques, Fabio Castro Gouveia, Elaine Teixeira Rabello, Gustavo Correa Matta, Mauricio L. Barreto, Ricardo Barros Sampaio.

**Writing – review & editing:** Juliane Fonseca de Oliveira, Julia Moreira Pescarini, Moreno de Souza Rodrigues, Bethania de Araujo Almeida, Claudio Maierovitch Pessanha Henriques, Fabio Castro Gouveia, Elaine Teixeira Rabello, Gustavo Correa Matta, Mauricio L. Barreto, Ricardo Barros Sampaio.

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
