## [Decision Letter · Decision Letter 0]

18 Sep 2019

PONE-D-19-19413

The global scientific research response to the public health emergency of Zika virus infection

PLOS ONE

Dear Dr Barros Sampaio,

Thank you for submitting your manuscript to PLOS ONE. After careful consideration, we feel that it has merit but does not fully meet PLOS ONE’s publication criteria as it currently stands. Therefore, we invite you to submit a revised version of the manuscript that addresses the points raised during the review process.

As you will gather from the reviewer comments, the data presented in your manuscript are particularly interesting and offer a wide and global vision on the scientific efforts that were made during the last ZIKV epidemic. I also express this view after careful examination of your study, which I find well suited for the journal. However, both the reviewer and myself found that a (final) section discussing what can be learned from the general conclusions of your data and the preparation for futur epidemic is currently lacking and needed. Specifically, even if poorly provided in data, a discussion on what was perceived by the community as not or poorly effective during the epidemic (*e.g*. Funding availabilities, timing of international programs, data collecting...) would greatly benefit your manuscript.  

Moreover, we note that your search was performed using only one database; to ensure that a thorough search is presented, we would request that you update the search to include at least a second database.

We would appreciate receiving your revised manuscript by December 2019. To enhance the reproducibility of your results, we recommend that if applicable you deposit your laboratory protocols in protocols.io, where a protocol can be assigned its own identifier (DOI) such that it can be cited independently in the future. For instructions see: http://journals.plos.org/plosone/s/submission-guidelines#loc-laboratory-protocols

We look forward to receiving your revised manuscript.

Kind regards,

Carmen Melatti, PhD, Staff Editor 

on behalf of:

Sara Salinas, PhD

Academic Editor

PLOS ONE

Journal Requirements:

[This work is supported by the Center of Data and Knowledge Integration for Health (CIDACS)

482 through the Zika Platform - a long-term surveillance platform for the Zika virus and

483 microcephaly, Unified Health System (SUS) - Brazilian Ministry of Health.

484

485 This work was partially supported by the European Union’s Horizon 2020 Research and

486 Innovation Programme under ZIKAlliance Grant Agreement no. 734548, and by the Oswaldo

487 Cruz Foundation/ Vice-Presidency of Research and Biological Collections-Fiocruz/ VPPCB and

488 the Newton Fund/ British Council.]

 [The author(s) received no specific funding for this work]

Reviewers' comments:

Reviewer's Responses to Questions

**Comments to the Author**

1. Is the manuscript technically sound, and do the data support the conclusions?

Reviewer #1: Yes

2. Has the statistical analysis been performed appropriately and rigorously? 

Reviewer #1: Yes

3. Have the authors made all data underlying the findings in their manuscript fully available?

Reviewer #1: No

4. Is the manuscript presented in an intelligible fashion and written in standard English?

Reviewer #1: Yes

5. Review Comments to the Author

Reviewer #1: In this study, authors aimed to verify how the Zika epidemic has molded scientific production worldwide analyzing international collaboration and the knowledge landscape through time, research topics and country involvement. Mapping the response to Zika, a public health emergency, demonstrated a clear pattern of the participation of countries in the scientific advances. The pattern of knowledge production found in this study represented the different perspectives and interests of countries based firstly on their level of exposure to the epidemic and secondly on their financial positions with regard to

science.

The data presented in this study are interesting. They confirm already known results: the leading role of the USA in biomedical research, the strong involvement of countries directly affected, the need to establish collaborations for more basic research. This work is very descriptive. It does not determine what has worked well and what has not worked in terms of international research and collaboration. This greatly limits the scope of this study. Indeed, we do not see what lessons could be learned for a future epidemic of this type.

6. PLOS authors have the option to publish the peer review history of their article (what does this mean?). If published, this will include your full peer review and any attached files.

Reviewer #1: No

---

## [Author Response · Author response to Decision Letter 0]

21 Jan 2020

Dear Editor and Reviewers,

With this letter, we resubmit the paper entitled “The global scientific research response to the public health emergency of Zika virus infection” (PONE-D-19-19413) for review to PLOS One after incorporating the comments from the reviewer and editor. The text has been fully revised taking into account the comments and suggestions made by the reviewer and editor. We answered all questions and incorporate all the suggestions, which we believe has made the manuscript more clear and understandable. We hope that you will now find it suitable for publication. Our responses to the reviewers appear on the pages that follow.

We thank you again for the opportunity to submit our revised manuscript for publication consideration. We believe that through addressing the reviewers’ comments, the quality of our paper has improved considerably. 

Yours Sincerely,

Ricardo Barros Sampaio

On behalf of all authors. 

1.1: However, both the reviewer and myself found that a (final) section discussing what can be learned from the general conclusions of your data and the preparation for future epidemic is currently lacking and needed. Specifically, even if poorly provided in data, a discussion on what was perceived by the community as not or poorly effective during the epidemic (e.g. Funding availabilities, timing of international programs, data collecting...) would greatly benefit your manuscript. 

Reply 1.1: Agreed. Discussion and Conclusion sections have been modified to incorporate how our findings can contribute to future epidemic responses and preparedness. Also, we improved the Introduction section aiming to clarify our objectives. 

1.2: Moreover, we note that your search was performed using only one database; to ensure that a thorough search is presented, we would request that you update the search to include at least a second database.

Reply 1.2: We thank the editor for this comment. In order to account for your suggestion, we performed searches in two additional databases: Scopus and PubMed. Descriptive analysis on the databases WoS, Scopus and PubMed was incorporated in the text and knowledge maps of words extracted from titles and abstracts of articles on WoS, Scopus and PubMed are presented in supplementary figures S1 Fig to S3 Fig. They were obtained using VOSviewer software. We can see that the clusters found from VOSviewer software for the three databases are very similar. This results support our choice to carry out textual analysis only with the WoS database.

Comment 1: Have the authors made all data underlying the findings in their manuscript fully available?

Reviewer #1: No

Reply 1: Thank you for this remark. We have included the data extracted from Wos, Scopus and Pubmed and the additional files created to conduct our analysis as Supplementary Material (S2 File). A description of the data used in the manuscript is presented in the S1 File.

2.1: It does not determine what has worked well and what has not worked in terms of international research and collaboration. This greatly limits the scope of this study. 

Reply 2.1: We believe that the success of the scientific collaboration is hard to measure but the utility of each manuscript, on the other hand, can be partially estimated by its citations. Nevertheless, in this manuscript, due to the limited time frame in which Zika research is available, we opted for not evaluating citation metrics. Please, see also the Reply 1.1 for the editor comment. 

2.2: Indeed, we do not see what lessons could be learned for a future epidemic of this type.

Reply 2.2: Please, see Reply 1.1 for the editor comment.

---

## [Decision Letter · Decision Letter 1]

14 Feb 2020

The global scientific research response to the public health emergency of Zika virus infection

PONE-D-19-19413R1

Dear Dr. Barros Sampaio, 

We are pleased to inform you that your manuscript has been judged scientifically suitable for publication and will be formally accepted for publication once it complies with all outstanding technical requirements.

With kind regards,

Sara Salinas, PhD

Academic Editor

PLOS ONE

Additional Editor Comments (optional):

Reviewers' comments:

Reviewer's Responses to Questions

**Comments to the Author**

1. If the authors have adequately addressed your comments raised in a previous round of review and you feel that this manuscript is now acceptable for publication, you may indicate that here to bypass the “Comments to the Author” section, enter your conflict of interest statement in the “Confidential to Editor” section, and submit your "Accept" recommendation.

Reviewer #1: All comments have been addressed

2. Is the manuscript technically sound, and do the data support the conclusions?

Reviewer #1: Yes

3. Has the statistical analysis been performed appropriately and rigorously? 

Reviewer #1: N/A

4. Have the authors made all data underlying the findings in their manuscript fully available?

Reviewer #1: Yes

5. Is the manuscript presented in an intelligible fashion and written in standard English?

Reviewer #1: Yes

6. Review Comments to the Author

Reviewer #1: The authors modified the manuscript and answered most of the questions. This has improved the quality of the study. This work deserves to be published in the journal.

7. PLOS authors have the option to publish the peer review history of their article (what does this mean?). If published, this will include your full peer review and any attached files.

Reviewer #1: No

---

## [Editor Report · Acceptance letter]

24 Feb 2020

PONE-D-19-19413R1 

The global scientific research response to the public health emergency of Zika virus infection 

Dear Dr. Sampaio:

I am pleased to inform you that your manuscript has been deemed suitable for publication in PLOS ONE. Congratulations! Your manuscript is now with our production department. 

With kind regards,

on behalf of

Dr. Sara Salinas 

Academic Editor

PLOS ONE